# Emergence of Two Distinct SARS-CoV-2 Gamma Variants and the Rapid Spread of P.1-like-II SARS-CoV-2 during the Second Wave of COVID-19 in Santa Catarina, Southern Brazil

**DOI:** 10.3390/v14040695

**Published:** 2022-03-27

**Authors:** Dayane Azevedo Padilha, Vilmar Benetti Filho, Renato Simões Moreira, Tatiany Aparecida Teixeira Soratto, Guilherme Augusto Maia, Ana Paula Christoff, Fernando Hartmann Barazzetti, Marcos André Schörner, Fernanda Luiza Ferrari, Carolina Leite Martins, Eric Kazuo Kawagoe, Julia Kinetz Wachter, Paula Sachet, Antuani Rafael Baptistella, Aline Daiane Schlindwein, Bruna Kellet Coelho, Sandra Bianchini Fernandes, Darcita Buerger Rovaris, Marlei Pickler Debiasi dos Anjos, Fernanda Rosene Melo, Bianca Bittencourt, Sthefani Cunha, Karine Lena Meneghetti, Nestor Wendt, Tâmela Zamboni Madaloz, Marcus Vinícius Duarte Rodrigues, Doris Sobral Marques Souza, Milene Höehr de Moraes, Rodrigo de Paula Baptista, Guilherme Toledo-Silva, Guilherme Razzera, Edmundo Carlos Grisard, Patricia Hermes Stoco, Luiz Felipe Valter de Oliveira, Maria Luiza Bazzo, Gislaine Fongaro, Glauber Wagner

**Affiliations:** 1Universidade Federal de Santa Catarina, Florianópolis 88040-900, Brazil; dayufsc@gmail.com (D.A.P.); vilmarbf98@gmail.com (V.B.F.); tsoratto@gmail.com (T.A.T.S.); guiaugmaia@gmail.com (G.A.M.); fernandohb55@hotmail.com (F.H.B.); marcos.schorner@gmail.com (M.A.S.); ferrari.fernandal@gmail.com (F.L.F.); carol4952@hotmail.com (C.L.M.); kazuo.eric@gmail.com (E.K.K.); kinetzjulia@gmail.com (J.K.W.); nestor.wendt@ufsc.br (N.W.); tamelamadaloz96@gmail.com (T.Z.M.); mrviniduarte@gmail.com (M.V.D.R.); doris.sobral@gmail.com (D.S.M.S.); guilherme.toledo@ufsc.br (G.T.-S.); guilherme.razzera@ufsc.br (G.R.); edmundo.grisard@ufsc.br (E.C.G.); patricia.stoco@ufsc.br (P.H.S.); marialuizabazzo@gmail.com (M.L.B.); 2Instituto Federal de Santa Catarina, Lages 88506-400, Brazil; renatosm@gmail.com; 3Biome-Hub Pesquisa e Desenvolvimento, Florianópolis 88054-700, Brazil; anachff@biome-hub.com (A.P.C.); milene.hoehr@biome-hub.com (M.H.d.M.); felipe@biome-hub.com (L.F.V.d.O.); 4Universidade de São Paulo, São Paulo 05508-220, Brazil; pssachet@gmail.com; 5Universidade do Oeste de Santa Catarina, Joaçaba 89600-000, Brazil; antuani.baptistella@unoesc.edu.br; 6Secretaria de Estado da Saúde, Florianópolis 88015-130, Brazil; alineds10@yahoo.com.br; 7Laboratório Central do Estado da Saúde de Santa Catarina, Florianópolis 88010-001, Brazil; brunakellet@gmail.com (B.K.C.); genomicalacensc@gmail.com (S.B.F.); darcitarovaris@gmail.com (D.B.R.); marleipickler@saude.sc.gov.br (M.P.D.d.A.); 8Diretoria de Vigilância Epidemiológica de Santa Catarina, Florianópolis 88015-130, Brazil; fernandar.melo@gmail.com (F.R.M.); bittenka@gmail.com (B.B.); 9Serviço Nacional de Aprendizagem Industrial, Chapecó 70040-903, Brazil; sthefani.cunha@sc.senai.br (S.C.); karine.meneghetti@sc.senai.br (K.L.M.); 10University of Georgia, Athens, GA 30602, USA; rodrigopbaptista@gmail.com

**Keywords:** SARS-CoV-2, surveillance, genome, SARS-CoV-2 P.1 variant

## Abstract

The western mesoregion of the state of Santa Catarina (SC), Southern Brazil, was heavily affected as a whole by the COVID-19 pandemic in early 2021. This study aimed to evaluate the dynamics of the SARS-CoV-2 virus spreading patterns in the SC state from March 2020 to April 2021 using genomic surveillance. During this period, there were 23 distinct variants, including Beta and Gamma, among which the Gamma and related lineages were predominant in the second pandemic wave within SC. A regionalization of P.1-like-II in the Western SC region was observed, concomitant to the increase in cases, mortality, and the case fatality rate (CFR) index. This is the first evidence of the regionalization of the SARS-CoV-2 transmission in SC and it highlights the importance of tracking the variants, dispersion, and impact of SARS-CoV-2 on the public health systems.

## 1. Introduction

The COVID-19 pandemic, caused by the SARS-CoV-2 virus, is the world’s most significant public health challenge of the last 100 years. This virus has infected more than 270 million people, leading to more than 5.3 million deaths worldwide by December 2021 [1]. In the same period, since the first reported case on December 2019, this virus infected around 22.2 million people, leading to 617,000 deaths in Brazil and in the southern state of Santa Catarina (SC), 1,238,056 cases and 20,101 deaths were reported [2].

With the virus’s rapid accumulation of polymorphisms and the advancement of the pandemic, new variants have emerged [3]. Thus, genomic surveillance has been essential for monitoring this disease and vital for local health agencies to have information to mitigate the effects of the pandemic and adopt strategies to increase population engagement in COVID control policies [4].

Until December 2021, more than 6.1 million SARS-CoV-2 genomes have been sequenced, allowing for the detection of several variants [5]. Some new variants have distinctive biological characteristics from their predecessors, such as better transmission, immunological escape or distinguishing clinical symptoms [6]. Some variants of SARS-CoV-2 have been defined as variants of interest (VOIs) or variants of concern (VOCs), with potential impact on public health. The main VOCs described have been B.1.1.7/VOC 202012 (Alpha), December 2020; B.1.351/501Y.V2 (Beta), December 2020; P.1/B.1.1.248/B1.1.28 (Gamma), January 2021; B.1.617.2 (Delta), May 2021 [1]. Recently, the B.1.1.529 (Omicron) variant was reported in South Africa in November 2021 [1], evidencing that new VOCs of SARS-CoV-2 are still emerging and, thus, genomic surveillance of the virus remains crucial.

In Brazil, the VOC Gamma (P.1) variant was first reported in the city of Manaus, state of Amazonas, in November 2020, being identified in 42% of RT-PCR positive samples collected between 15th and 23rd December 2020 [7]. This VOC is 1.7–2.6 times more transmissible than previous strains circulating in Brazil, showing a 1.2–1.9-fold increase in the mortality risk to adults and 5–8-fold for the young population [8], broadening the susceptible population under risk of infection and hospitalization as well as the needs of hospitalization and intensive care [9].

Among several mutations for VOC Gamma, the E484K mutation occurs in the receptor-binding domain (RBD) domain of the S protein and is believed to assist the evasion of the virus from the immune system [10] and, therefore, may lead to lower vaccine efficacy [11]. Additionally, present in the RBD domain, the N501Y mutation is associated with increased binding specificity to the host cell angiotensin-converting enzyme 2 (ACE2) receptor. Subvariants of VOC Gamma and related lineages (P.1-like-I and P.1-like-II), displaying distinct mutations, also emerged during 2021 [12]. Among these, P.1.1, P.1.2, and P.1-like-II (Gamma-like-II) spread across the border states of Rio Grande do Sul [13] and Paraná [14] until April 2021 and were concomitantly detected in SC state [15].

In this context, our study evaluated the dispersion of different variants of SARS-CoV-2 from 1st March 2020 to 30th April 2021 in the state of Santa Catarina, especially during the second wave of the pandemic (January to April 2021) when the VOC Gamma variant became predominant in Brazil.

## 2. Material and Methods

### 2.1. Studied Regions and Sampling

The state of Santa Catarina (SC) is located in the southern region of Brazil, having borders with the Paraná (north) and Rio Grande do Sul (south) states, and Argentina (west). With an estimated population of 7,333,473 inhabitants, the SC state is geographically divided into six mesoregions—Greater Florianópolis, Itajaí River Valley, Mountain range, Northern, Southern, and Western SC [16].

Genome sequences from 203 SARS-CoV-2 positive samples collected from different mesoregions of the state of Santa Catarina, Brazil from 1st March 2020 to 30th April 2021, were generated (Figure 1) (Appendix A). Additionally, 527 SARS-CoV-2 genome sequences obtained from samples isolated in SC (Figure 1) and 49 genome sequences from samples isolated worldwide, including the original SARS-CoV-2 sequence (NC_045512.2), were retrieved from the EpiCov database GISAID [5] and included in this study (only genome sequences showing <5% ambiguous nucleotide and >85% genome sequence coverage). This study was approved by the Federal University of Santa Catarina Ethics Committee (CAAE: 31521920.8.0000.0121).

### 2.2. RNA Purification and Sequencing

The study used 203 human nasopharyngeal swab samples from SARS-CoV-2 positive patients diagnosed via RT-qPCR. The inclusion criterion was a positive RT-PCR for gene E with C_t_ values of <25. The total viral RNA was extracted using the RNA Viral Kit (QIAGEN, Hilden, Germany), and the long amplicon pooled protocol was used [17]. Briefly, complementary DNA (cDNA) was synthesized from the viral RNA using 50 μM random primers and 200 U/μL Superscript IV reverse transcriptase (ThermoFisher Scientific, Waltham, MA, USA). The obtained cDNA was amplified using 10 μM of 14 primer sets in individual reactions to amplify different regions of the SARS-CoV-2 genome (approximately 2.5 Kb in each fragment) and 40 U/μL Platinum *Taq* DNA polymerase High Fidelity (Invitrogen, Waltham, MA, USA). Amplicons were visualized on agarose 2% gel electrophoresis.

The amplicons were then pooled, purified using the AMPure XP (Beckman Coulter, Brea, CA, USA), and quantified by Quant-iT Picogreen dsDNA assays (Invitrogen, Waltham, MA, USA). The pools were then processed using the Illumina DNA Prep (Illumina, San Diego, CA, USA), following the manufacturer’s protocol. Libraries were quantified using the Collibri Library Quantification kit (ThermoFisher Scientific, Waltham, MA, USA), adjusted to 11.5 pM and prepared for paired-end sequencing (2 × 150 bp) using the MiSeq Reagent Kit V2 (300-cycles) (Illumina, San Diego, CA, USA). Sequencing was performed on the Illumina MiSeq platform (Illumina, San Diego, CA, USA).

### 2.3. SARS-CoV-2 Genome Assembling and Variant Analysis

The SARS-CoV-2 genome was assembled according to the ARTIC protocol [18]. Briefly, low-quality bases (Q < 25) and Illumina adapters were trimmed [19] and mapped to the human reference genome (hg19—GCF_000001405.13) to remove contaminants using BWA v. 0.7.17 [20]. Next, human-free reads were aligned to the SARS-CoV-2 reference genome (NC_045512.2) using BWA with default parameters. Ivar v. 1.3 [21] and the mpileup function from samtools v. 1.7 [22] were used to perform variant calling and to generate genome consensus sequences. The assembled genomes were analyzed in the Nextclade [23] and Pangolin [24] platforms to classify the SARS-CoV-2 variants.

Mutations, deletions, and insertions were used to assess the dissimilarity of the genomes using the vegan R package v. 2.5.6 [25] with the Jaccard index method. The multidimensional scaling (MDS) and the distribution of variants in SC were plotted using the ggplot2 R package v. 3.3.0 [26].

### 2.4. Maximum-Likelihood (ML) Phylogenetic Analysis

The ML phylogeny was performed using all 779 SARS-CoV-2 genome sequences (Appendix A). Sequences were aligned by MAFFT v. 7.310 [27] with default parameters and were manually curated using AliView v.1.26 [28]. An ML tree was constructed using the alignment, with GTR + F + I as the substitution model predicted by ModelFinder [29], 1000 bootstrap runs using UFBoot tree optimization [30] and SH-like approximate likelihood ratio test (SH-aLRT) [31], using IQ-TREE v. 2.1.3 [32]. The ML tree was visualized and annotated using the ggtree R package v. 2.0.4 [33].

### 2.5. Network Analysis of SC VOC Gamma and Related Lineages during the Second Wave

The network analysis was performed using 589 Brazilian VOC Gamma and related genomes retrieved from GISAID, of which 418 were SC SAR-CoV-2 genomes from samples obtained from 5th January 2021 to 30th April 2021, and 171 genome sequences were from other Brazilian states sequenced up to 5th January 2021 (Appendix A). The ML consensus tree was constructed as above, except the substitution model was GTR + F + R2. The network analysis was performed using the online service StrainHub with closeness centrality [34], the ML consensus tree, and corresponding metadata.

### 2.6. Spike (S) Gene Sequence Polymorphism and Protein Structure Analysis

For structural analysis of the S protein of VOC Gamma, 57 presented full-length S protein coverage of VOC Gamma sequenced in this study, with 722 retrieved from GISAID (3 December 2021), including the reference sequence (NC_045512.2), were aligned using MAFFT [27]. The mutations were represented over the 3D S protein structure PDB 6ZGG (resolution 3.8 Å). The experimental structure presents a representative coverage of the N-terminal domain (NTD) (Gln14 to Asp1146). The mutations were highlighted and colored according to the observed frequencies and built with PyMOL version 2.3.3 software [35].

## 3. Results

### 3.1. Profile of SARS-CoV-2 Variants in the State of Santa Catarina after One Year of the COVID-19 Pandemic

A total of 23 distinct SARS-CoV-2 variants were identified among the studied genomes (Figure 2A). Of these 730 complete genomes, 316 (43.2%) were classified as the VOC Gamma, followed by 175 (23.9%) of the former VOI Zeta variant (P.2) and 69 (9.4%) of the P1-like II variant. Only four sequences of the VOC Alpha variant (B.1.1.7) were observed among the analyzed samples.

The first sequenced sample was classified as a B.39 variant. Then, the B.1.1.28 and the B.1.1.33 variants prevailed from May to November 2020, when the P.2 variant gradually became the most prevalent. The P.2 variant peaked in December 2020 and was displaced quickly by the P.1 variant. By January 2021, the VOC Gamma had already taken over most of the samples in Santa Catarina (Figure 2A). Until December 2020, the profiles of the variants were similar among the mesoregions. However, in the beginning of the second wave (February 2021), the western mesoregion had a distinct profile, with the highest cases/100,000 inhabitants indices (Appendix A) and the P.1-like-II lineage being predominant in that region (Figure 2B). The Mountain Range was sampled only in February 2021, where 10 VOC Gamma and a single P.1-like-II were detected.

### 3.2. VOC Gamma and Related Lineages in Santa Catarina

We inferred phylogenetic relationships between the SARS-CoV-2 genomes from SC and the other known variants (Figure 3). Most of the SC sequences were grouped in the VOC Gamma clade, which includes subvariants P.1.1, P.1.2, and P.1.10, and related lineages (P.1-like I and P.1-like II). However, this parent clade generates two distinct sub-clades, one containing the variant P.1-like-II and P.1-like-I, and the other clustering sequences classified as P.1 and their subvariants. It is worth mentioning that the clade formed mainly by P.1-like-I and P.1-like-II variants is composed of samples from Western SC, pointing out a possible independent introduction and regionalized transmission of such variants in that region.

A multidimensional scale (MDS) plot shows the clustering of the SARS-CoV-2 genomes according to the mutations observed (Figure 4A). There was no clear clustering of the studied SARS-CoV-2 variants according to the sampled geographic regions along the sampling period (Figure 4B). However, in a separate analysis of the data from the VOC Gamma and the related lineages, it was possible to observe a clustering of genomes from Western SC (Figure 4C), mainly composed of the P.1-like-II variants (Figure 4D).

As the ML and the MDS analyses pointed out the evidence of the genetic proximity of the cluster of P.1-related lineage (P.1-like-II) in Western SC, we compared all SC VOC Gamma and related lineage sequences obtained from 7th August 2020 to 30th April 2021. These genomes were classified based on the amino acid substitutions described in P.1-like-I [36] and P.1-like-II [13] variants. Mutations that characterize each VOC Gamma-related lineage profile were identified and are summarized in Figure 5.

More than 90% of the S proteins of the VOC Gamma and related lineages present the mutation L18F, P26S, D138Y, and R190S positions of the NTD region (Figure 6). The typical P.1 mutations were found in the RBD domain (K417T, E484K, and N501Y), SD domain (D614G and H655Y), and overall structure (T1027I, and V1176F). The original amino acid T20 was conserved in the NTD in 20% of the sequences, being characterized as P.1-like-II lineages.

### 3.3. The Rapid Spread and Regionalization of P.1-like-II during the Second Wave of the COVID-19 Pandemic in Santa Catarina, Brazil

During the second wave of COVID-19 (January–April 2021) the VOC Gamma and subvariants were the most represented in SC. However, there was not a homogeneous variant distribution pattern in the mesoregions (Figure 2B). Network analysis revealed that VOC Gamma and related lineage genomes had genetic proximity with genomes sequenced in the state of Amazonas (Figure 7A). The VOC Gamma and related lineages from Western SC had fewer interactions with other SC mesoregions than with the state of Amazonas (Figure 7B). The city of Chapecó, SC, a major municipality of Western Santa Catarina, acted as a point of concentration and dissemination of lineages imported from Manaus to other western cities (Figure 7C).

In addition, it was observed that in February and March 2021, 75% and 60% of the sequences in Western SC were classified as P.1-like-II, respectively (Figure 8). Furthermore, in the same period, this mesoregion had the highest case fatality rate (CFR), mortality, and number of deaths in the state of Santa Catarina (Figure 9), which indicates a rapid dissemination of the P.1-like-II variant in the region, followed by an increase in deaths.

Taken together, these data indicate distinct introductions of VOC Gamma and related lineages in Santa Catarina, followed by a rapid dispersion of the P.1-like-II variant in the western region of SC and a regionalization in the transmission of P.1-like-II in the western region.

## 4. Discussion

In this study carried out between March 2020–April 2021, we sequenced 203 SARS-CoV-2 genomes from human samples collected from different regions of the SC state, representing 27.8% of the virus genomic sequences generated for this region. The assessment of the SARS-CoV-2 genetic profiles (Figure 2A) allowed for the identification of VOC Alpha (20I/501Y.V1/B.1.1.7) and especially the Gamma (20J/501Y.V3/P.1) from January 2021 onwards, corresponding to the second wave of the pandemic in SC, as also observed in other Brazilian states [37,38] and indicating the homogeneity of the spreading pattern within Brazil at that time. However, between August 2020 and November 2020, we were able to identify the emergence of the formerly VOI P.2 [1] in SC, which rapidly turned out to be more prevalent than the other variants circulating in the region. This variant was originally identified in Brazil in Rio de Janeiro by July 2020, although phylogenetic analyses indicated that the variant may have arisen as early as February 2020 [39]. Similar to our findings in SC, P.2 was reported to have become predominant in several Brazilian states around the same period, especially in northeastern and southeastern Brazil, when variant P.1 turned out to be more prevalent by the beginning of 2021 [39], except for the state of Amazonas, where the P.2 variant was first detected in November 2020 but was overshadowed by P.1 since its early detection and, thus, never became the main circulating variant [38]. After the emergence of VOC Gamma in Manaus City, state of Amazonas [7], the first case of P.1 in SC was reported in Florianópolis City (Greater Florianópolis) (hCoV-19/Brazil/SC-1326/2021) by 8th January 2021, and the first case of VOC Gamma-related lineage (P.1-like-II) in Chapecó City (Western Santa Catarina) on 22nd January 2021, the latter also reported by Gräff et al. [12]. The genetic variability of SARS-CoV-2 observed in our study is shown by the variants found here (Figure 2), which probably emerged because of the genome replication during the viral infection. Synthesis of RNA virus particles can produce an average of 100,000 viral copies in approximately 10 h, with one RNA molecule produced every 0.4 s and without a repair system of this new RNA produced [40,41]. ML analysis showed that P.1-like-II samples obtained from patients living in the western mesoregion of SC were clustered into a single, separate clade (Figure 3), while viral samples from the other SC regions due to scarce P.1-like-II were mostly VOC Gamma and subvariants, pointing out that the introduction and initial spreading of the P.1-like-II variant in SC occurred via the western mesoregion (Figure 4C,D). In addition, as highlighted in Figure 5 and Appendix A, VOC Gamma is characterized by three mutations in the receptor-binding domain (RBD)—K417T, E484K, and N501Y—that can modulate ACE2/RBD affinity, increasing transmissibility or even the affinity of the antibody. In addition to the mutations L18F, T20N, P26S, D138Y, and R190S in the N-terminus (NTD), D614G and H655Y in the C-terminus of S1, and T1027I and V1176F in S2 were also reported in VOC Gamma [42]. All these mutations were identified in SC; however, clusters 3 and 8 containing the P.1-like-II lineage are formed by sequences with the absence of a T20N mutation and are composed especially of samples from the western region (Figure 5). The appearance of P.1-like-II, as well as several of the variants sequenced here, is a phenomenon of cumulative mutations characteristic of RNA viruses. The accumulation of SARS-CoV-2 genome mutations may be related to transmissibility, host cell affinity, and pathogenicity [43]. These characteristics may explain what was observed in Figure 8—that the fatalities were higher in the western region of the state of Santa Catarina, where the P.1-Like-II variant was in greater proportion.

The evolution of RNA viruses is based on amino acid substitution, the replacement rate of SARS-CoV-2 being very similar to other RNA viruses, such as Influenza A and human enterovirus, and the replacement rate for SARS-CoV-2 is 1–3 replacements per site per year, approximately two replacements per month, for circulating viruses [44,45]. Amino acid insertions are common in spike proteins (Figure 5 and Appendix A), which may be a natural process of viral adaptation. Investigating changes in structural S proteins is of paramount importance, given the need for these proteins in virus–host cell binding [46].

Thus, in this study, the analysis of such markers allowed for the identification of two main lineages that are distinguished by the presence or absence of the T20N mutation in the S protein. This SARS-CoV-2 protein is highly glycosylated, being involved in the virus evasion mechanisms of the human immune system through mutations that hinder viral recognition by the immune system. Most of these glycosylation sites are located in Asp residues that contain N-glycosylation sites, or Ser and Thr residues that contain O-glycosylation sites. Among these, the most frequent glycosylation sites were found in N + 2, where Asp is followed by Ser/Thr residues [47]. Remarkably, the herein reported T20N mutation at the NTD is followed by a Thr (N + 2) in all analyzed samples (N20X21T22) and introduces a new potential O-glycosylation site within the S protein that may increase the virus capability of evading the immune defenses of viral neutralization by means of antibodies, and not altering the potential N-glycosylation site at N17 that is also present in P.1 variants, where a Thr follows an Asp residue at position 19 (N17X18T19).

As shown in Figure 4D, by February 2021, we identified a high frequency of the subvariant P.1-like-II in samples from Western SC compared to the other regions of the state, preceding an increase in the CFR index, mortality rate, and deaths (Figure 9) in that mesoregion. Taken together, these data indicate that P.1-like-II had high transmissibility and effect on the lethality in Western SC. Naveca et al. [38] showed how the second wave of COVID-19 in the state of Amazonas, with a sudden rise in cases and number of deaths during late 2020 and early 2021, directly correlates to the rise in Gamma frequency in the state, which had reached a massive prevalence by January 2021. Faria et al. [7] demonstrated that the infection by VOC Gamma is 1.1–1.8 times more severe than those previously observed by other variants, considering that the results of this variant were related to higher viral loads, greater transmissibility, and an increase in the number of deaths.

The relationship between the increase in the number of cases and the appearance of variants was also observed outside of Brazil, such as in India and Europe. In India, the appearance of the Delta variant was observed during the second wave, correlating with the consequent increase in the number of deaths. In Europe, the second wave was also marked by an increase in the number of cases and deaths, which correlated with the appearance of the Alpha variant [48,49]. However, the transmission rate of SARS-CoV-2 may be affected by a variety of factors, and the emergence of a given variant could not be considered the unique factor to increase such rates [50]. Among several virus-inherent factors, such as genetic variability, the lack or misuse of non-pharmaceutical interventions (NPI), social difficulties for diagnosis or late diagnosis, uncontrolled population movement [51], the collapse of the public health system [52], lower vaccination [53], the overtime decrease of the immune protection [52], late or poor decisions made by authorities [54], and the use of scientifically unproven preventive therapies [55,56,57,58], there are also important factors that might contribute to the increases in the transmission and case fatality rates. Examples in different countries show that an increase in the movement of people and low widespread adherence to the NPI was observed weeks before the collapse of the public health service, and the consequent increase in the number of cases and deaths [59,60]. Indeed, the SARS-CoV-2 circulation was evidenced by the detection of viral RNA copies in sewage months before being reported in people, as had occurred in Santa Catarina state in November 2019 [61]. Thus, we cannot rule out that the conjunction of both viral and other biological, environmental, and behavioral factors have significant influence on the occurrence of SARS-CoV-2 variants presenting variable infectivity and pathological effects. Such factors may have contributed to the differentially increased number of cases, hospitalization, and deaths (CRF) in a short period of time in Western SC, as also previously observed in Manaus (Amazon, Brazil), having totally distinct environmental conditions from SC.

With the following report of the VOC Omicron in South Africa in November 2021 [1], this variant was rapidly detected worldwide [1] and reported in SC, at the beginning of December 2021, evidencing the ongoing viral adaptations and the continuing pandemic spreading pattern. Considering these facts, the need for the maintenance of a detailed and standardized genomic surveillance of SARS-CoV-2 variants towards precise and specific identification of VOCs is of utmost importance, either by high-throughput sequencing, as herein proposed, or nucleic acid amplification-based tests (NAAT) [62,63,64], as a faster, widely used, and less time-consuming alternative, avoiding erroneous epidemiological inferences and allowing for the evaluation of vaccine efficacy across viral variants.

Thus, high-throughput sequencing-based genomic surveillance of the SARS-CoV-2 pandemic has been crucial to assess the accumulation of mutations, mostly due to the enormous quantity of viral replication events, and the biological outcomes of the genetic variability of each variant on the epidemiology, transmissibility, infectivity, pathogenesis, and vaccine efficacy and effectiveness. Since major concerns over the genetic evolution of SARS-CoV-2 are related to an increased spreading capability and to an escape from the immune system, genomic surveillance provides essential data to prevent the current pandemic from lasting even longer.

## 5. Conclusions

This study allowed us to observe the distinct patterns of circulation of different SARS-CoV-2 variants within SC regions. Our data suggest a distinct introduction and rapid spreading of P.1-like-II in the western regions of SC in the initial months of 2021, where it was responsible for distinct and increased hospitalization and mortality rates. Thus, the continuous real-time genomic surveillance of SARS-CoV-2 in Santa Catarina can offer precise epidemiological support for the implementation and review of policies and actions carried out by public health authorities towards the control of COVID-19 transmission and the assessment of immunization efficacy.

## Disclaimer

The opinions expressed by authors contributing to this journal do not necessarily reflect the opinions of the Federal University of Santa Catarina, Brazil, or the institutions with which the authors are affiliated. The funders had no role in the study design, data analysis, or the decision to publish.

## Figures and Tables

**Figure 1 viruses-14-00695-f001:**
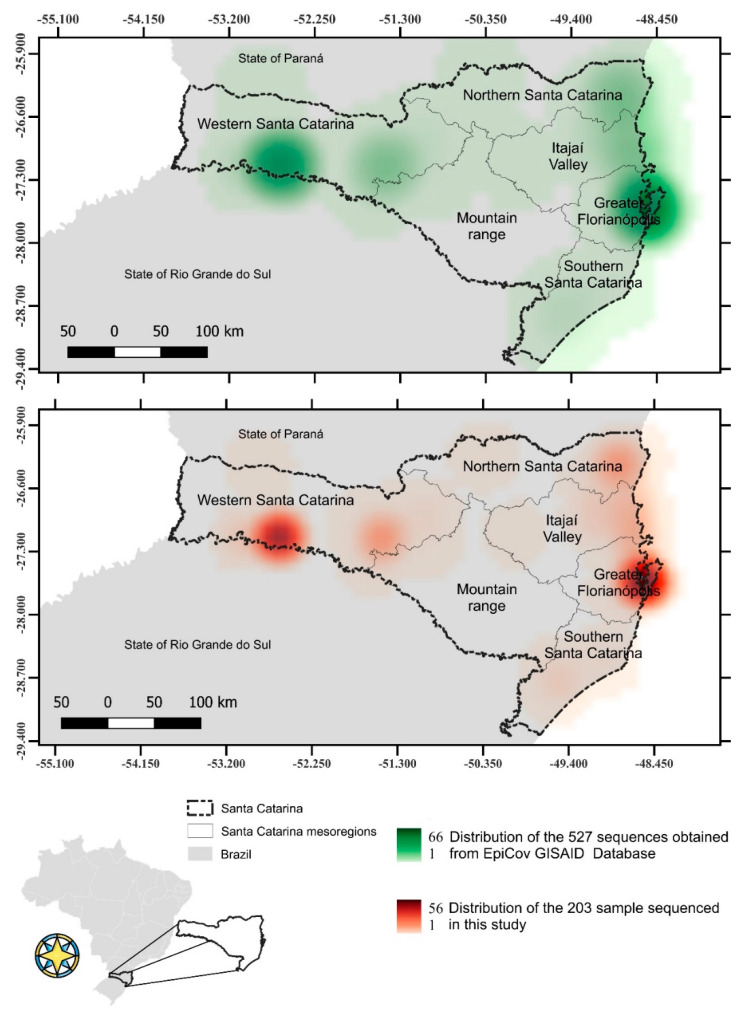
Distribution of sequenced SARS-CoV-2 genomes in Santa Catarina from 1st March 2020 to 30th April 2021. Map showing the Santa Catarina SARS-CoV-2 sequences obtained from GISAD EpiCov Database (upper map) and samples sequenced in this study (lower map).

**Figure 2 viruses-14-00695-f002:**
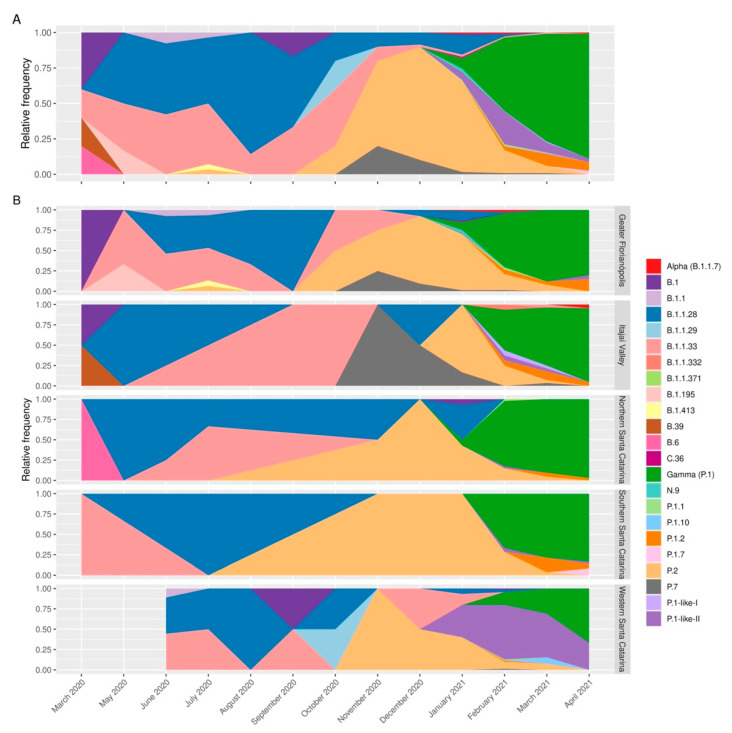
SARS-CoV-2 variant profiles in the state of Santa Catarina from March 2020 to April 2021. (**A**) Analysis of the variant profile considering the whole state of Santa Catarina. (**B**) The variant profile by Santa Catarina mesoregion. From March to September 2020, the variants identified were in the highest proportion of B.1.1.28 (blue) and B.1.1.33 (light red). From September 2020 to January 2021, the highest proportion was of P.2 variant, followed by VOC Gamma and related lineages from February to April 2021.

**Figure 3 viruses-14-00695-f003:**
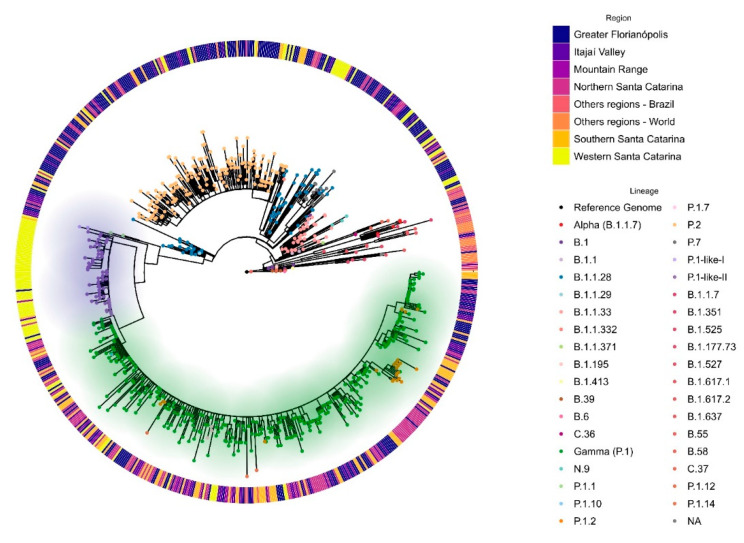
Phylogenomic reconstruction analysis of Santa Catarina SARS-CoV-2 sequences from 1st March 2020 to 30th April 2021. Maximum likelihood (ML) phylogenetic tree was reconstructed using GTR + F + I substitution model, with 1000 iterations in Ultra-Fast Bootstrap mode with SH-like approximate likelihood ratio test (SH-aLRT). All 779 sequences were aligned with 49 reference sequences of distinct SARS-CoV-2 variants available in GISAID. Dots represent the variants or lineages indicated on the right-bottom legend. According to the right-top legend, the outside bars indicate the sample mesoregion of Santa Catarina or others. The green shadow indicates the VOC Gamma and related lineages (P.1.1, P1.2, P.1.10, and P.1.7) clade. Lilac shadow indicates the P.1-like-II clade.

**Figure 4 viruses-14-00695-f004:**
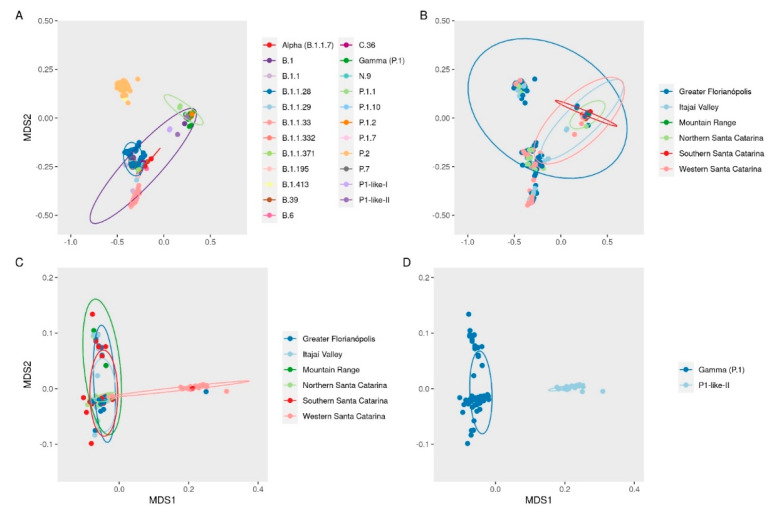
Multidimensional scale (MDS) plot of Jaccard dissimilarity matrix between genomes based on the presence and absence of mutations, deletions, and insertions labeled by (**A**) variants and (**B**) region. MDS analysis of only VOC Gamma and related lineages labeled by (**C**) region and (**D**) variant or lineages.

**Figure 5 viruses-14-00695-f005:**
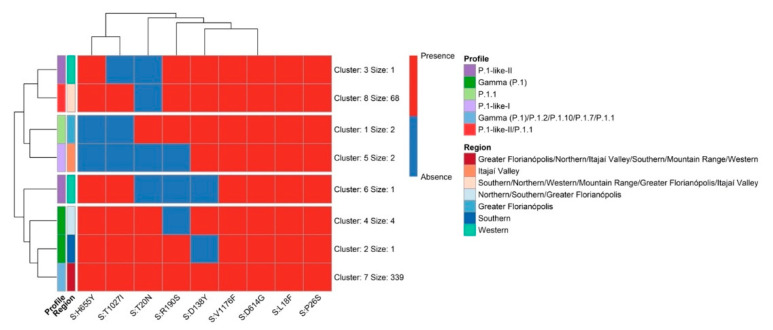
The non-synonymous amino acid substitution profile of the VOC Gamma and related lineages genomes in the state of Santa Catarina from 3 January 2020 to 30 April 2021.

**Figure 6 viruses-14-00695-f006:**
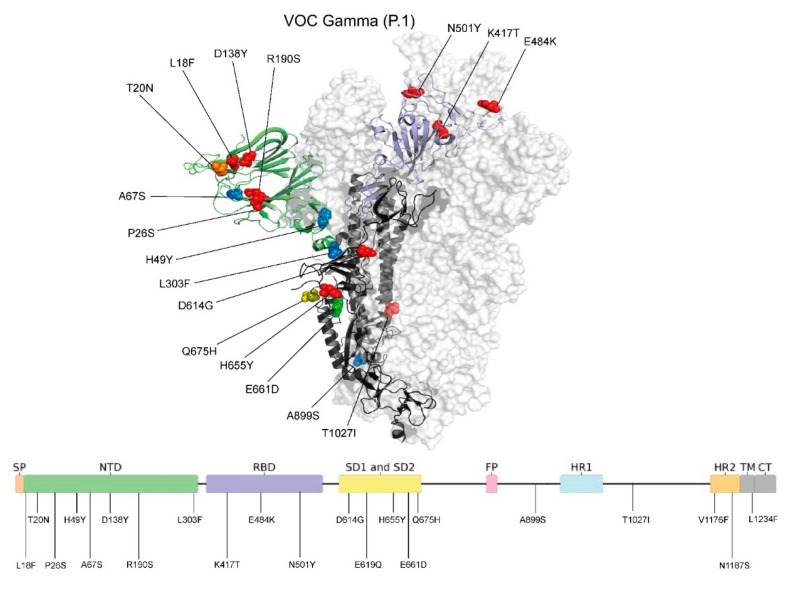
Overlay of VOC Gamma spike protein mutations identified in the state of Santa Catarina during the COVID-19 pandemic. Residues involved in mutations are presented as spheres, colored from blue to red (low to high mutation frequency). The mutation sites are also indicated in the two-dimensional representation of the spike protein, which is colored according to protein regions. The NTD domain is colored in green and the RBD domain is colored in purple. From the total 778 analyzed sequences of the VOC Gamma variant (5′–3′), red represents a frequency of 93.6–100%, orange a frequency of 88.8%, yellow a frequency of 3.98%, green a frequency of 1.79%, and blue a frequency of less than 1%. Image generated using PyMOL version 2.3.3 [35].

**Figure 7 viruses-14-00695-f007:**
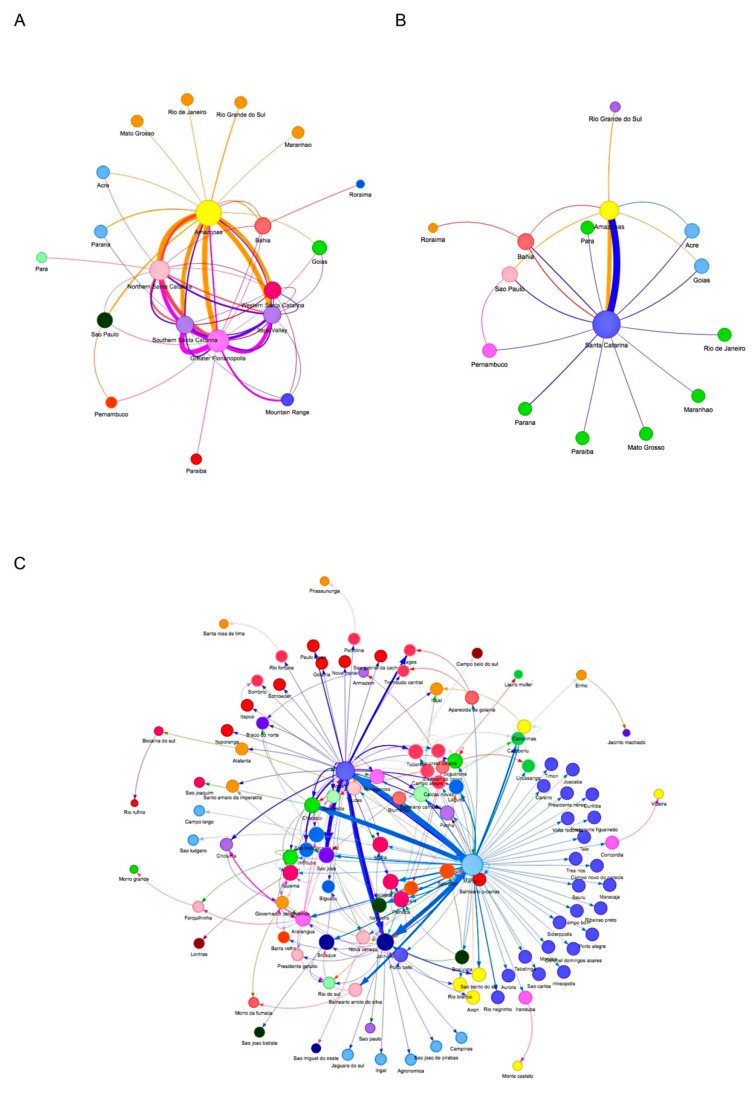
Network analysis of VOC Gamma and related lineages sequenced in the state of Santa Catarina, Brazil, from 3rd January 2020 to 30th April 2021 with the other Brazilian states. The ML analysis was performed on 13,018 Brazilian VOC Gamma-related genomes obtained from GISAID until 30th April 2021 and submitted to StrainHub [34] analysis considering the closeness parameter. The sizes of the nodes are scaled by the closeness metric and the arrows reflect the directionality of possible transmissions among the Brazilian states (**A**), Santa Catarina mesoregions (**B**), and Santa Catarina cities (**C**). The thickness of the lines and arrows represents the frequency of putative transmissions.

**Figure 8 viruses-14-00695-f008:**
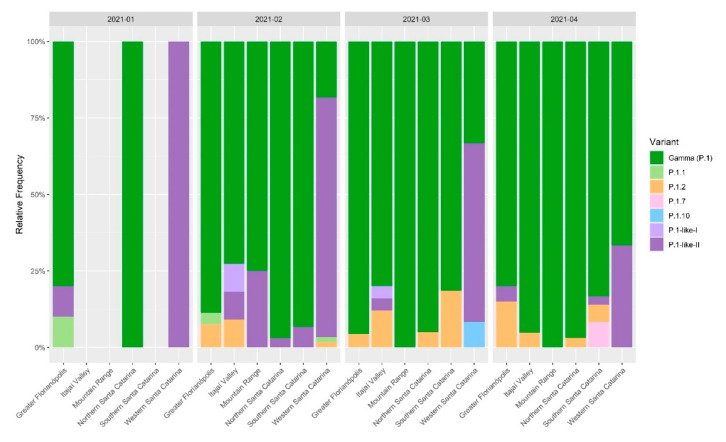
Relative frequencies of the SARS-CoV-2 VOC Gamma and related lineages by mesoregions of the state of Santa Catarina during the second wave of the COVID-19 pandemic, January to April 2021. A representative alignment of spike proteins from these sequences is available in Appendix A.

**Figure 9 viruses-14-00695-f009:**
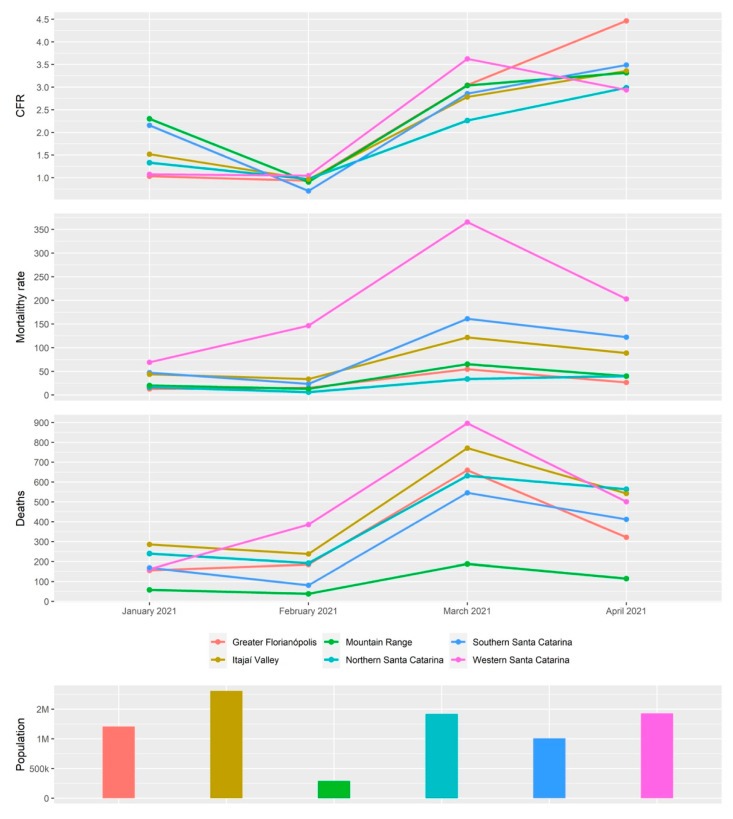
Epidemiological data from each mesoregion of the state of Santa Catarina during the second COVID-19 pandemic wave (January 2020 to April 2021). Case fatality rate (CFR), mortality rate, number of deaths, and population size. Data source: Santa Catarina Epidemiological Surveillance Directorate [15].

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
