# Peer review of "Emergence of Two Distinct SARS-CoV-2 Gamma Variants and the Rapid Spread of P.1-like-II SARS-CoV-2 during the Second Wave of COVID-19 in Santa Catarina, Southern Brazil"

_viruses, 2022, doi:10.3390/v14040695_

Round 1

Reviewer 1 Report

Comments for authors

The manuscript submitted by Dayane etal., ‘’Emergence of two distinct SARS-CoV-2 Gamma variants and the rapid spread of P.1-like-II SARS-CoV-2 during the second wave of COVID-19 in Santa Catarina, Southern Brazil” is a very interesting study. Authors have evaluated the dynamics of SAR-CoV-2 transmission by sequencing the viral genomes from human samples that was collected from different regions of the Santa Catarina state in Brazil. Overall, the manuscript is well written, and the data is well presented. I have only minor comments for the authors.

  1. I think authors must include the aminoacid sequence alignment for full length or portion of spike protein for the representative VOC Gamma and related lineages that was listed in the Figure 7. Having this information, it will be helpful for the readers to learn what are the differences between them.

  1. Supplementary tables are missing.

  1. I find the importance of those mutations that characterize the VOC Gamma related lineage profile. I think Supplementary figure 2 should be moved to the main figures.

Author Response

The manuscript submitted by Dayane et al., ‘’Emergence of two distinct SARS-CoV-2 Gamma variants and the rapid spread of P.1-like-II SARS-CoV-2 during the second wave of COVID-19 in Santa Catarina, Southern Brazil” is a very interesting study. Authors have evaluated the dynamics of SAR-CoV-2 transmission by sequencing the viral genomes from human samples that was collected from different regions of the Santa Catarina state in Brazil. Overall, the manuscript is well written, and the data is well presented. I have only minor comments for the authors.

Answer: We appreciated the reviewer comment.

Comment I: “I think authors must include the aminoacid sequence alignment for full length or portion of spike protein for the representative VOC Gamma and related lineages that was listed in the Figure 7. Having this information, it will be helpful for the readers to learn what are the differences between them.”

Answer: We appreciated the reviewer comment. We agree with the reviewer's suggestion and include in the revised manuscript a representative alignment figure (Logo) of VOC Gamma sequences and related strains (Page 12, Figure 8 legend). In this alignment is possible to observe the non-synonymous substitution T20N in the Spike protein that allows to differentiate the VOC Gamma sequences from the P.1-like-II. We also included the complete Spike protein alignment at the end on this, however because of figure size, we chose to include as a supplementary figure only the representative alignment.

Comment II: Supplementary tables are missing.

Answer: We appreciated the reviewer comment. The supplementary tables were reviewed and submitted correctly.

Comment III: I find the importance of those mutations that characterize the VOC Gamma related lineage profile. I think Supplementary figure 2 should be moved to the main figures.

Answer: We agree with the reviewer's suggestion and supplementary figure 2 was included as a figure in the revised manuscript, which is now Figure 5. All references of the figures in the main text and the subtitles were updated in the revised manuscript.

Reviewer 2 Report

See attached file

Author Response

The authors utilized genome sequencing of confirmed COVID-19 cases to map the spread of SARS-CoV-2 variants during the second wave of COVID-19 in the Santa Catarina region of Brazil. These results indicated that the Gamma variant was the most prominent circulating strain during the second wave and also indicates possible increased mortality associated with the P.1-like-II variant localized in the western region. It appears that the P.1-like-II lineage arose from variant lacking the T20N mutation found in most other Gamma strains. It is hypothesized that this P.1-like-II variant was introduced in the western region based on the distribution. These results follow trends from other regions of Brazil and provide useful data about how the pandemic progressed in different regions of the world.

We appreciated the reviewer comment.

Comment I: “Strengths: The authors do an appropriate of job of limiting their surveillance to a designated area and analyzing the results within that context. They were able to analyze approximately ¼ of the total positive cases to represent an appropriate sample size. The figures (maps, profile analysis, phylogeny, modeling) are well presented and increase understanding of the data being presented. The supplementary figures are also well presented and help to add visual understanding of the spread and prevalence. The discussion includes information about potential glycosylation impacts, data correlation with other regions of Brazil, and potential social impacts on fatality rates which helps to temper the correlations identified in the surveillance. Overall, the paper is well written and needs only minimum edits for English and grammar.”

Answer: We appreciated the reviewer comment. The English and grammar were revised. 

Comment II: “Weaknesses: The paper suffers from natural weaknesses in the scope of the sampled area, but the authors appropriately address their conclusions based on this limitation. While the majority of the figures are of a high production quality, figure 8 does stand out as being lower quality, perhaps consider improving the look and overall quality of this figure. The P.1-like-II variant has also been referred to as the “Gamma-like-II” variant, consider including this name when first introducing the P.1-like-II variant.”.

Answer: We appreciated the reviewer comment. The term Gamma-like-II was included on the first time that P.1-like-II (Page 2).

Comment III: Introduction First paragraph – Deaths are listed as “617 thousand”, change to “617,000” for continuity with rest of paper.

Answer: We appreciated the reviewer recommendation, and it was changed in the revised manuscript (Page 2)

Comment IV: Results First paragraph – delete “in” following Zeta variant (P.2) 

Answer: We appreciated the reviewer recommendation, and it was removed in the revised manuscript (Page 5)

Reviewer 3 Report

The Authors present a study in which they  evaluated the dispersion of different variants of SARS-CoV-2 through March 1st, 2020 to April 30th, 2021, in the state of Santa Catarina, using NGS approach. The study is weel written and the results are presented and discussed in exhaustive way. Anyway, the discussion could be implemented with a brief description concerning  possible alternative methods (screening methods), cheaper and more rapid than NGS. Such screening methods (be in-house methods as well as commercial assay) generally using a RT-PCR approach. Parcticularly, in low- or mid-income countries, they could be useful in the rapid screening and evaluation of the SARS-CoV-2 variant circulation.

Author Response

Comment I: The Authors present a study in which they evaluated the dispersion of different variants of SARS-CoV-2 through March 1st, 2020 to April 30th, 2021, in the state of Santa Catarina, using NGS approach. The study is weel written and the results are presented and discussed in exhaustive way. Anyway, the discussion could be implemented with a brief description concerning possible alternative methods (screening methods), cheaper and more rapid than NGS. Such screening methods (be in-house methods as well as commercial assay) generally using a RT-PCR approach. Particularly, in low- or mid-income countries, they could be useful in the rapid screening and evaluation of the SARS-CoV-2 variant circulation.

Answer: We acknowledge the reviewer comment. The suggested information was included (Page 16 and new references).